# Ex Situ Synthesis and Characterizations of MoS_2_/WO_3_ Heterostructures for Efficient Photocatalytic Degradation of RhB

**DOI:** 10.3390/nano12172974

**Published:** 2022-08-28

**Authors:** Wajeehah Shahid, Faryal Idrees, Muhammad Aamir Iqbal, Muhammad Umair Tariq, Samiah Shahid, Jeong Ryeol Choi

**Affiliations:** 1Department of Physics, University of Lahore, Lahore 54000, Pakistan; 2Department of Physics, University of the Punjab, Lahore 54590, Pakistan; 3School of Materials Science and Engineering, Zhejiang University, Hangzhou 310027, China; 4Institute of Molecular Biology and Biotechnology, University of Lahore, Lahore 54000, Pakistan; 5Department of Nanoengineering, Kyonggi University, Suwon 16227, Korea

**Keywords:** MoS_2_/WO_3_, heterostructures, solar light, photocatalysis, degradation

## Abstract

In this study, novel hydrothermal ex situ synthesis was adopted to synthesize MoS_2_/WO_3_ heterostructures using two different molar ratios of 1:1 and 1:4. The “bottom-up” assembly was successfully developed to synthesize spherical and flaky-shaped heterostructures. Their structural, morphological, compositional, and bandgap characterizations were investigated through XRD, EDX, SEM, UV-Visible spectroscopy, and FTIR analysis. These analyses help to understand the agglomerated heterostructures of MoS_2_/WO_3_ for their possible photocatalytic application. Therefore, prepared heterostructures were tested for RhB photodegradation using solar light irradiation. The % efficiency of MoS_2_/WO_3_ composites for 30 min irradiation of 1:1 was 91.41% and for 1:4 was 98.16%. Similarly, the % efficiency of 1:1 MoS_2_/WO_3_ heterostructures for 60 min exposure was 92.68%; for 1:4, it was observed as 98.56%; and for 90 min exposure, the % efficiency of 1:1 was 92.41%, and 98.48% was calculated for 1:4 composites. The photocatalytic efficiency was further verified by reusability experiments (three cycles), and the characterization results afterward indicated the ensemble of crystalline planes that were responsible for the high efficiency. Moreover, these heterostructures showed stability over three cycles, indicating their future applications for other photocatalytic applications.

## 1. Introduction

The unprecedented growth of industrialization has led to organic contaminants in the environment. The degradation of pollutants by using semiconductor photocatalysis has been regarded as a promising technology [1,2]. Photocatalysis has gained increasing attention recently due to its potential to provide an ecologically friendly method for converting sunlight into chemical energy in mild reaction environments through photocatalytic processes and photochemical reactions. Several heterojunction photocatalysts have been developed and are being employed for water-splitting and the photocatalytic degradation of organic compounds. Among them, solar-responsive metallic oxides/sulfides have gained attention due to their effective redox processes [3,4].

For visible light photocatalysis, tungsten trioxide (WO_3_) is an excellent choice because of its nontoxicity and distinctive optical properties. In addition, it is inexpensive, easy to synthesize, has strong stability in both acidic and basic environments, and has high electron carrier efficiency [5]. Due to its limited light absorption range and the quick recombination of photo-generated electron-hole pairs, pure WO_3_ photocatalytic activity is severely limited. To improve solar light absorption, impurities and higher conduction band semiconducting materials such as TiO_2_ (−0.29 eV), Ag_3_PO_4_ (+0.3 eV), and BiVO_4_ (0 eV) were used [6]. The addition of impurities by doping, on the other hand, reduces the thermal stability of the material and provides defect sites that act as additional recombination centers. A unique approach for separating photo-generated electron–hole pairs driven by a self-built electric field while also broadening the spectrum absorption range is currently being investigated [5].

Molybdenum disulfide (MoS_2_) is a widely used photocatalyst due to its large specific surface area, abundant unsaturated active sites, and visible spectrum absorption. The photochemical performance of MoS_2_ can be improved by coupling it with other semiconductors [6]. MoS_2_ possesses a more powerful negative conduction band (−0.06 eV) than WO_3_; the addition of MoS_2_ might make it possible to overcome WO_3_’s low conduction band position. Few studies have reported on the use of MoS_2_ on WO_3_ to increase photocatalytic performance because of the very small contact area between these two semiconductors [7,8].

Contrary to the single-component system, heterojunction photocatalysts have become a more feasible option for the breakdown of toxic contaminants. Here, the ex situ approach was used to synthesize binary composites of MoS_2_/WO_3_ in various weight ratios [9]. Combining two catalysts with similar band gaps increases photocatalytic activity by creating a heterojunction, which increases the promotion of photo-generated charges and decreases the recombination rate. Therefore, a heterojunction can be formed using several semiconducting materials, such as WO_3_/MoS_2_, WO_3_/CuBi_2_O_4_, and WO_3_/CdS, which would improve the catalyst effectiveness and ability to absorb light. Given that MoS_2_ has a sizable surface area and strong electrical mobility, the MoS_2_/WO_3_ (MSW) pair is an excellent choice for creating the heterojunction [10].

We have successfully synthesized WO_3_, MoS_2_, and MoS_2_/WO_3_ composites and evaluated photocatalytic degradation of Rhodamine (RhB) dye under solar light. A significant increase in the rate of photodegradation of RhB in the presence of heterojunction photocatalysts has been observed. To investigate the photocatalytic effectiveness of ex situ hydrothermally synthesized MoS_2_/WO_3_ nanocomposites of varied weight ratios, Rh B photodegradation studies were carried out in this study. Several investigations, including pH, concentration, and reusability assays, were conducted to evaluate the effectiveness and performance of the catalyst prepared. To demonstrate the photocatalyst’s superiority, its degrading effectiveness was also compared to that of the literature.

Additionally, the mechanism of the composite-enhanced photocatalytic activity was investigated, which could be useful for understanding the hydrogen evolution process for future applications.

## 2. Materials and Methods

All the chemicals used were of analytical grade, purchased from Sigma-Aldrich, (St. Louis, MO, USA), and were used without further purification.

### 2.1. Synthesis of MoS_2_

MoS_2_ was produced utilizing a two-step hydrothermal technique under acidic circumstances, employing MoO_3_ and Ammonium Thiocyanate (NH_4_SCN) as starting materials. A total of 1.5 mmol MoO_3_ (0.22 g) was dissolved in 40 mL of deionized water and sonicated for 30 min, after which the pH value of the solution was adjusted to 1 by stirring for 30 min with a 1 mol/L HCl solution. These samples were then transferred to a 50 mL Teflon-lined stainless steel autoclave for hydrothermal treatment at 180 °C for 12 h, which was cooled down at room temperature, yielding a black powder of MoS_2_ by centrifugation for 10–15 min, washed multiple times with ethanol and deionized water, and dried for 12 h in an oven at 80 °C.

### 2.2. Synthesis of WO_3_

The sodium tungstate dehydrates (Na_2_WO_4_) (0.1 M) were dissolved in distilled water (10 mL) to maintain a pH value of ~8. Then, hydrochloric acid (HCl) (0.5 M) was added dropwise at 50 °C to form a homogeneous solution. After continuous stirring for 15 min, the pH of the resulting solution was set to ~1. The resulting solution was then shifted to a 50 mL Teflon-lined Stainless-Steel Autoclave and placed in the oven at 180 °C for 6 h for hydrothermal treatment. After naturally cooling down, blue precipitates were separated, centrifuged, and washed several times simultaneously with deionized water and absolute ethanol, then dried in an oven at 80 °C for 12 h. The resultant powder was obtained after annealing at 400 °C for 2 h in a muffle furnace.

### 2.3. Synthesis of MoS_2_/WO_3_

Already-prepared MoS_2_ and WO_3_ samples were used for this process. MoS_2_/WO_3_ (1:1) and MoS_2_/WO_3_ (1:4) nanocomposites were prepared by using an ex situ synthesis in which the first MoS_2_ solution was prepared with 10 mL of ethanol, and the second WO_3_ solution was prepared with 10 mL of ethanol. Following that, WO_3_ solution was added dropwise to the MoS_2_ solution while continuous stirring was maintained for the next 30 min. The resulting solution was then centrifuged and washed simultaneously with deionized water and ethanol. The resultant dark greyish powder was obtained after drying in an oven at 80 °C for 12 h. After drying, the agglomerated material was ground in mortar and pestle to a fine powder of MoS_2_/WO_3_ (1:1) and MoS_2_/WO_3_ (1:4).

### 2.4. Photocatalytic Activity

The RhB dye was used to investigate photocatalytic activity. The research was carried out under the presence of a solar simulator, namely, the “Abet Technologies Sunlight TM Solar Simulator.” The catalysts (MoS_2_/WO_3_) were washed by a series of centrifugation and washing steps before being reused for the subsequent degradation. Three cycles were performed to test the catalysts’ reusability and stability. The dye solution catalytic decolorization is a pseudo-first-order reaction, and the degradation rate was estimated using Equation (1) [11]:(1)$$\mathrm{Degradation}\;\left(\%\right)=\left(1-C_{t}/C_{0}\right)\times 100\%$$
where $C_{t}$ and $C_{0}$ are the dye’s concentration at time *t* and initial concentration, respectively. The degradation of 100 mL of aqueous RhB dye (10 ppm) was examined at dark, 30 min, 60 min, and 90 min. All the tests were carried out with 0.1 mg of the catalyst at its natural pH. During the photocatalytic degradation process, the distinctive absorption peak of RhB at 554 nm was set. In the absence of a catalyst, there was essentially no degradation, or at the very least, the rate of degradation was minimal. The catalytic activity of MoS_2_/WO_3_ was also tested in the dark for one hour.

## 3. Results and Discussion

### 3.1. Structural Analysis

Structural properties of MoS_2_/WO_3_ heterostructures were examined out using XRD analysis, as shown in Figure 1. The XRD was performed through Analytical X’Pert emitting CuK (Alpha) X-rays at the scanning range of 10–70°. All the diffracted peaks for MoS_2_/WO_3_-(1:1) and MoS_2_/WO_3_-(1:4) prepared heterostructures were matched with JCPDS 73-1508 for MoS_2_ and JCPDS 89-4480 for WO_3_, confirming the successful synthesis of heterostructures. The corresponding hkl parameters are marked in Figure 1. The observed pattern shows the increase in the WO_3_ ratio, and the intensity of WO_3_ peaks is also increased compared to MoS_2_ peaks. Moreover, the composition of the MoS_2_/WO_3_ heterostructures was also confirmed by using the EDX spectra, as shown in Appendix A.

### 3.2. Morphological Analysis

SEM analysis of MoS_2_/WO_3_ heterostructures (SEM) is shown in Figure 2a–d. Figure 2a,b are SEM images of MoS_2_/WO_3_ (1:4) heterostructures, where the prepared nanostructures have a relatively small size in comparison to Figure 2c,d for MoS_2_/WO_3_ (1:1). Figure 2b spherical agglomerates are smaller than Figure 2d agglomerates due to changed concentration, which may result in a decrease in crystalline size and an increase in the surface area [12]. SEM images of MoS_2_ and WO_3_ has been provided in Appendix A.

### 3.3. FTIR Analysis

For WO_3_ (in Figure 3), a very weak shoulder peak observed at 930 cm^−1^ was attributed to the W=O stretching vibration. As strong absorption peaks of W-O vibrational frequency near 1100 cm^−1^ in MoS_2_/WO_3_ 1:1 and 1:4 heterostructures, this peak was slightly shifted to 1100 cm^−1^ [12], and it overlaps with W-O at 1100 cm^−1^ in MoS_2_/WO_3_ (1:1) and MoS_2_/WO_3_ (1:4) heterostructure peaks. Due to the overlap of S-O and W-O peaks, the intense sharp peaks of both heterostructures appear as stronger peaks than the parent WO_3_ and MoS_2_ molecules. Another peak at 1365 cm^−1^ was absent in WO_3_ but appeared as a prominent peak in MoS_2_/WO_3_ 1:1 and 1:4 heterostructures corresponding to W-OH vibrational frequency in the form of a bending peak, but slight and broad and intensely sharp peaks were observed at 3450 cm^−1^ and 1612 cm^−1^ corresponding to OH stretching and bending peaks due to water molecules present in the WO_3_ crystal, respectively. Both became broader, more intense, and sharper in MoS_2_/WO_3_ 1:1 and 1:4, which shows the presence of string H-bonding. A sharp peak of Mo-OH vibration was observed for both heterostructures spectra, but it was absent in the WO_3_ spectrum and appeared as a slight impression in the MoS_2_ spectrum. It is inferred that both heterostructures were prepared by strong forces, which help composite molecules bind firmly together, and H-bonding due to water crystallization plays a vital role in this regard [12,13].

### 3.4. UV-Visible Spectroscopy Analysis

The UV-Vis spectroscopy model Shimadzu UV-1800 (Kyoto, Japan) was used to record absorption. Furthermore, by using Tauc’s plot, the optical bandgaps of prepared heterostructures of MoS_2_/WO_3_ were calculated by using Equation (2) [14].
(2)$$\alpha h\vartheta =A\;{\left(h\vartheta -E_{g}\right)}^{n},$$
where *n* depends on electron transitions, and *n* = 1/2 corresponds to indirect electron transitions with an indirect optical bandgap, and where *hϑ* is the photon energy, *α* is a constant, and *E_g_* is the optical bandgap energy. MoS_2_/WO_3_ (1:1) exhibited a bandgap of 2.06 eV in Figure 4a, while MoS_2_/WO_3_ (1:4) exhibited a narrow bandgap of 1.59 eV in Figure 4b. Their absorption spectra have been provided in Appendix A.

### 3.5. RhB Photodegradation

Photocatalytic activity of the prepared MoS_2_/WO_3_ heterostructures at 1:1 was performed by studying the degradation of RhB under solar light irradiation. The absorption intensity peak of RhB was centered at 554 nm and gradually decreased with an increase in the irradiation time, as shown in Figure 5a. Figure 5b represents the absorption and photodegradation curves of an aqueous solution of RhB photodegraded by a photocatalyst at a concentration of 0.1 gm over time. Photodegradation of MoS_2_/WO_3_ heterostructures with 1:4 under dark conditions was also observed, and the characteristic peak was taken as 554 nm of RhB dye used as a standard parameter during the photocatalytic degradation process. It was also observed that the concentration of RhB decreased with time, which was attributed to its degradation by a catalyst [14].

The photocatalytic activity was observed under solar light irradiation for 30 min, 60 min, and 90 min in Figure 6a, showing the relationship between % efficiency and irradiation time for MoS_2_/WO_3_ (1:1) and (1:4) heterostructures. Efficiencies of 91.41%, 92.41%, and 92.68% were observed for MoS_2_/WO_3_ (1:1), and efficiencies of 98.16%, 98.48%, and 98.56% were observed for MoS_2_/WO_3_ (1:4).

Figure 6b shows the degradation rate (C/C_0_) versus time (minutes), i.e., under dark, it was 0.0448, while for 30 min of solar light irradiation, it was 0.0858; for 60 min, it was 0.7314, and for 90 min it was 0.0758 for MoS_2_/WO_3_ (1:1). Likewise, the degradation rate (C/C_0_) under dark for MoS_2_/WO_3_ (1:4) was observed as 1, for 30 min under solar light irradiation it was 0.018, for 60 min it was 0.014, and for 90 min it was 0.015.

Figure 6c shows the graph between Ln(C/C_0_) vs. time in minutes and vs. % efficiency. The values of Ln(C/C_0_) for MoS_2_/WO_3_ (1:1), for 30, 60, and 90 min are 2.454, 2.615, and 2.579, respectively. Meanwhile, MoS_2_/WO_3_ (1:4) Ln(C/C_0_) values for 30, 60, and 90 min are 3.996, 4.244, and 4.189, respectively. The rate constant k/h for MoS_2_/WO_3_ (1:1) is calculated as 0.0278 and 0.0425 MoS_2_/WO_3_ (1:4), indicating first-order kinetics with excellent photocatalytic activity. The stability of MoS_2_/WO_3_ prepared heterostructures was tested for up to three cycles, as shown in Figure 6d. After 90 min of solar irradiation, prepared heterostructures showed excellent degradation and stability.

### 3.6. Proposed Photocatalytic Mechanism

The MoS_2_/WO_3_ heterostructure demonstrated enhanced photocatalytic potential in two ways: (i) generation of p-n heterojunction by band alignment through the close contact interface and (ii) efficiency as a co-catalyst. Both mechanisms reduce incoming light by forming electron–hole pairs. Under thermodynamic conditions, electrons in the MoS_2_ conduction band (CB) migrate to WO_3_ due to its bigger negative Fermi level, leaving holes in MoS_2_. Electrons in MoS_2_ CB can partially react with O_2_ to generate •O_2_, and holes in WO3’s VB can partially react with H_2_O to form •OH. Due to MoS_2_ having slightly higher CB potential than O_2_/•O_2_, the electron’s reduction ability was so feeble that the production rate of •O_2_ was substantially slower. The VB potential of WO_3_ was larger than that of H_2_O/•OH; therefore, enough •OH could be formed. The active species, •O_2_ and OH, interacted with organic molecules, causing them to oxidize and produce CO_2_ and H_2_O as by-products. In addition, the holes acted as active specie directs, oxidizing the RhB to the final product. Figure 7 depicts the photocatalytic mechanism of MoS_2_/WO_3_ [15,16]. 

## 4. Conclusions

A heterostructure binary nanocomposite, MoS_2_/WO_3_, at 1:1 and 1:4 ratios, was successfully prepared via ex situ synthesis. These prepared heterostructures were characterized, and the formation of heterojunctions was confirmed through XRD analysis. The optical bandgap energies for the MoS_2_/WO_3_ heterostructure were calculated as 2.06 eV and 1.59 eV for 1:1 and 1:4, respectively. An efficient RhB photodegradation was observed under solar light irradiation for prepared heterostructures. Among the already reported photocatalysts, MoS_2_/WO_3_ prepared heterostructures showed enhanced photocatalytic activity. The maximum photodegradation efficiency for MoS_2_/WO_3_ (1:4) was observed as 98.5%, indicating it is a suitable material for efficient photocatalytic degradation. The value of the K/h rate constant was observed as 0.0425, indicating the best photocatalytic activity was found when using RhB dye at 1:4. The relation between irradiation time to the rate constant Ln(C/C_0_) for MoS_2_/WO_3_ heterostructures was observed as 0.0278 for MoS_2_/WO_3_ at 1:1. MoS_2_/WO_3_ showed a quick and overall degradation ability for RhB in just 30 min with good reusability behavior and photostability. With suitable bandgap engineering, the current heterostructure could be used for molecular hydrogen production. All of the mentioned findings indicate that the MoS_2_/WO_3_ composites made would have a wide range of potential applications for eliminating organic dyes from wastewater.

## Figures and Tables

**Figure 1 nanomaterials-12-02974-f001:**
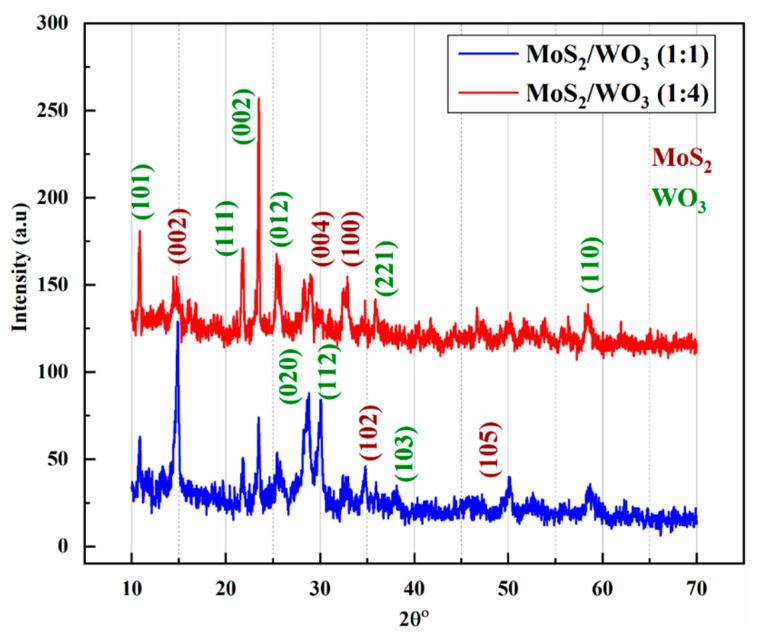
XRD patterns of MoS_2_/WO_3_ heterostructures; blue represents the molar ratio (1:1), while red shows (1:4).

**Figure 2 nanomaterials-12-02974-f002:**
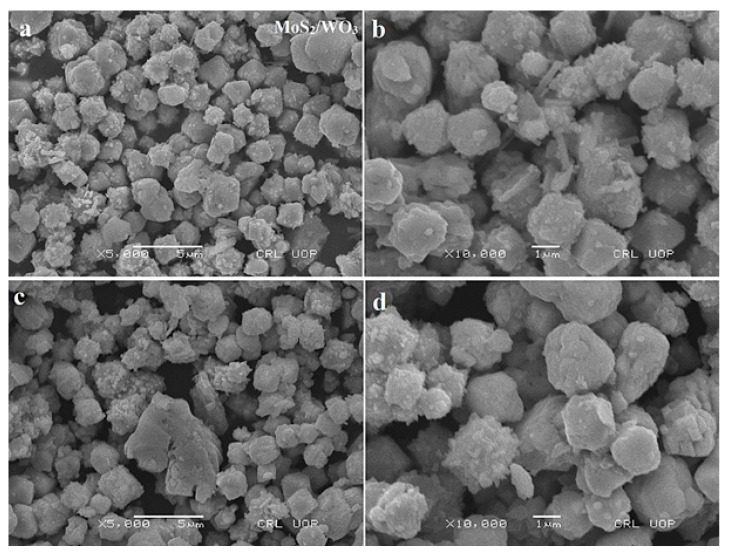
SEM Analysis of MoS_2_/WO_3_ with (**a**) molar ratio 1:4 at 5 µm, (**b**) molar ratio 1:4 at 1 µm, (**c**) molar ratio 1:1 at 5 µm, and (**d**) molar ratio 1:1 at 1 µm.

**Figure 3 nanomaterials-12-02974-f003:**
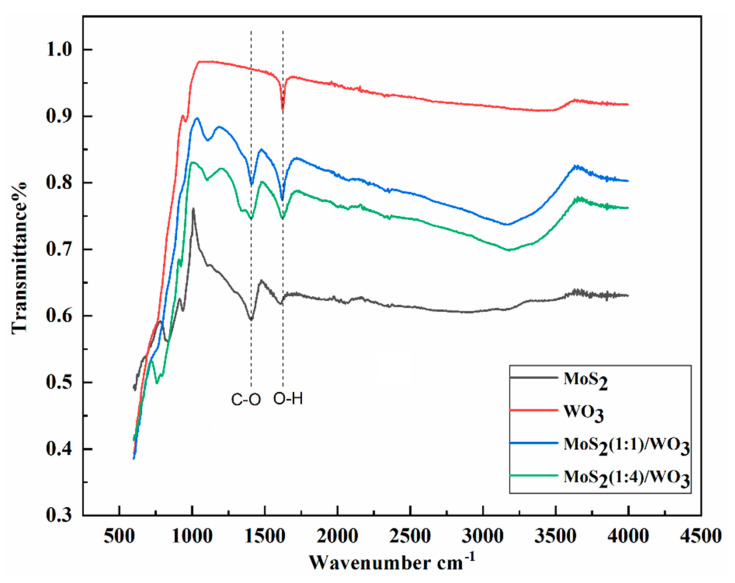
FTIR transmission spectra for (black) MoS_2_/WO_3_ (1:1), (red) MoS_2_/WO_3_ (1:4), (blue) MoS_2_, and (green) WO_3_.

**Figure 4 nanomaterials-12-02974-f004:**
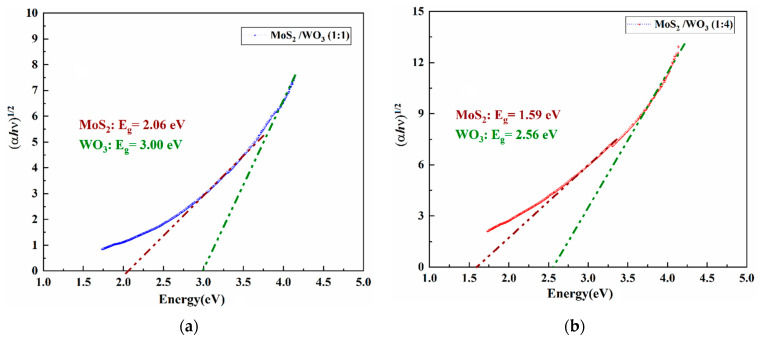
Optical bandgap of (**a**) MoS_2_/WO_3_ (1:1) and (**b**) MoS_2_/WO_3_ (1:4).

**Figure 5 nanomaterials-12-02974-f005:**
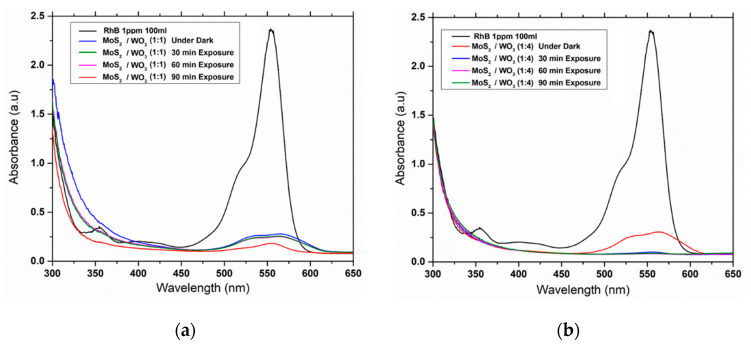
Graphs between wavelength vs. absorbance for MoS_2_/WO_3_ under dark conditions and at varying times of 30, 60, and 90 min for molar ratios: (**a**) (1:1) and (**b**) (1:4).

**Figure 6 nanomaterials-12-02974-f006:**
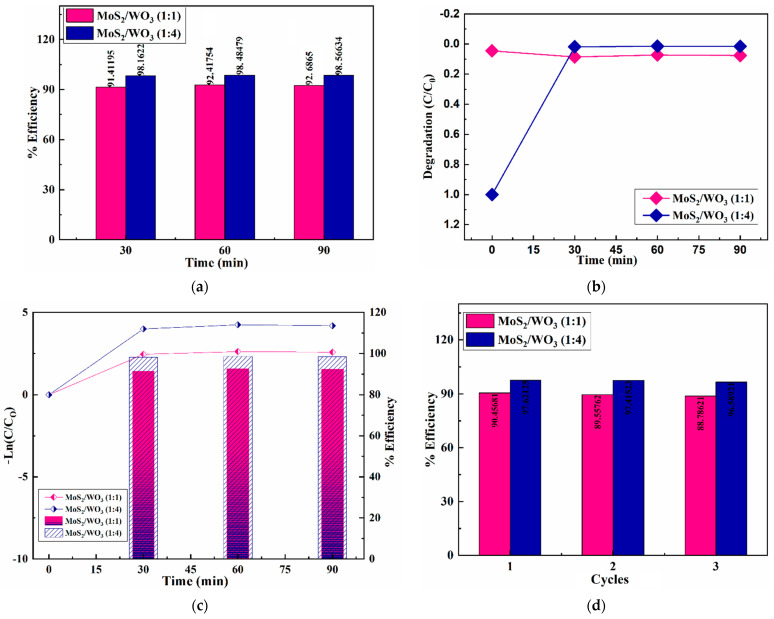
Graphs of: (**a**) % efficiency and time; (**b**) irradiation time vs. degradation (C/C_0_); (**c**) irradiation times versus Ln(C/C_0_) and % efficiency; (**d**) cycles vs. % efficiency for MoS_2_/WO_3_ prepared heterostructures.

**Figure 7 nanomaterials-12-02974-f007:**
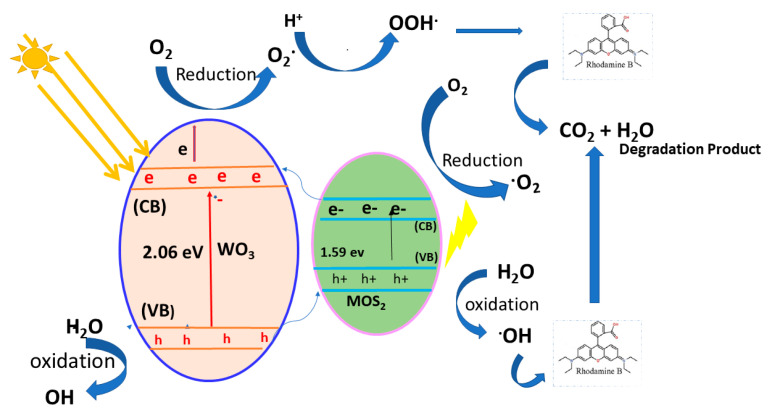
Schematic of photodegradation for MoS_2_/WO_3_.

## Data Availability

Not applicable.

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
