# Peer review of "Ex Situ Synthesis and Characterizations of MoS2/WO3 Heterostructures for Efficient Photocatalytic Degradation of RhB"

_nanomaterials, 2022, doi:10.3390/nano12172974_

Round 1

Reviewer 1 Report

1.       In the abstract, please add the several sentences to describe significance of the RhB detection and novelty.

2.       The introduction part is too cumbersome to highlight the innovative point and the main research content, which needs substantial modification.

3.       The place of procurement of chemicals/equipment must be mentioned. So, it is recommended that the author should include the instrumentation section in the revised paper.

4.       Provide the photoluminescence (Pl) and DRS of samples and calculate the net band gap, also surface area play important role to provide the BET in detail.

5.       Role of active radical species experiments needs to provide in photocatalytic performance and also focus on stability experiments of superior samples along with and acute toxicity estimation.

6.       Provide Possible photocatalytic degradation pathway of RhB dye along with its structure in supporting information. Compare your current results with previous reported results.

7.       Please focus to the following parameters such as Effect of catalyst load, Effect of initial concentration of dye, Adsorption phenomena of catalysts, Effect of solar light source and as well as PH and Kinetics of solar photocatalytic degradation of RhB. Standard addition technique should be applied for the determination of RhB in terms of validation.

8.       The conclusion section was poor. It should be improved with more analytical data.

9.       It’s is better to provide the digital photograph of samples to prove the color of samples.

10.   The English of this manuscript needs to be improved, such as making every sentence shorter and correct. Labelling every sample needs to be added and the figures caption contains some mistakes.

Author Response

Dear Reviewers and Editor,

            We would like to thank you for useful comments and suggestions on our manuscript that helped significantly in improving this manuscript. Your valuable suggestions have been incorporated in this version of the revised manuscript. The detailed modifications/changes have been listed below point by point and highlighted in the revised manuscript.

Reviewer #1:

  1. In the abstract, please add the several sentences to describe significance of the RhB detection and novelty.

Response: Thanks for your valuable suggestion. We have revised the abstract as per suggestion and has highlighted in the manuscript.

  1. The introduction part is too cumbersome to highlight the innovative point and the main research content, which needs substantial modification.

Response: Thanks, we have corrected the introduction as per your suggestions.

  1. The place of procurement of chemicals/equipment must be mentioned. So, it is recommended that the author should include the instrumentation section in the revised paper.

Response: All the chemicals were purchased from Sigma Aldrich (USA) included in instrumentation section.

  1. Provide the photoluminescence (Pl) and DRS of samples and calculate the net band gap, also surface area play important role to provide the BET in detail.

Response: Thanks for your valuable suggestion, SEM was used to determine the surface morphology, as samples are not porous therefore BET was not required. Moreover, these characterizations are not possible in this short time as we must ask other institution and it’ll takes time.

  1. Role of active radical species experiments needs to provide in photocatalytic performance and also focus on stability experiments of superior samples along with and acute toxicity estimation.

Response: Thanks, the schematic of stepwise degradation of RhB by the action of free radicals produce by photocatalyst has been incorporated in Figure 7 as well as provided in supporting information.

  1. Provide Possible photocatalytic degradation pathway of RhB dye along with its structure in supporting information. Compare your current results with previous reported results.

Response: Thanks for your valuable suggestion, incorporated in Figure 8 and a comparison with literature has been provided. Results are compared with literature which is highlighted in the manuscript in line number 235.

  1. Aa Please focus to the following parameters such as Effect of catalyst load, Effect of initial concentration of dye, Adsorption phenomena of catalysts, Effect of solar light source and as well as PH and Kinetics of solar photocatalytic degradation of RhB. Standard addition technique should be applied for the determination of RhB in terms of validation.

Response: Thanks for your valuable suggestion, Effect of the initial concentration of dye and PH has been provided in the experimental part which is highlighted in the manuscript. Kinetics of Solar photocatalytic degradation of RhB has been provided in Figure 5 of the manuscript.

  1. The conclusion section was poor. It should be improved with more analytical data.

Response: Thanks for your valuable suggestion it has been improved accordingly.

  1. It’s is better to provide the digital photograph of samples to prove the color of samples.

Response: Incorporated in the Figure S5 of the supporting information.

  1. The English of this manuscript needs to be improved, such as making every sentence shorter and correct. Labelling every sample needs to be added and the figures caption contains some mistakes.

Response: Thanks for your valuable suggestion. Extensive language editing has been done and the manuscript has been improved thoroughly.

Reviewer 2 Report

The paper describes a range of MoS2/WO3 photocatalysts that can efficiently degrade RhB under sunlight, and the motivation behind the problem studied in this manuscript is of interest. Nevertheless, there are a number of issues in this paper that need to be reconsidered after major revision. Mydetailedcommentsareasfollows:

1.     The “Ex-situ Synthesis” in the title is not reflected in the whole article. If this is not the highlight of this article, we suggest deleting it.

2.    The title of the article and the Materials and Methods section both indicate that the experiments in this paper were conducted under “Solar Light”, and it’s easy to make people confused that whether it’s real environmental sunlight or simulated sunlight. If it is real environmental sunlight, the light will be affected by weather or other conditions and therefore whether there is any control during the experiment. If not, it’d better to change the word o avoid confusion.

3.   The statement that the product has excellent catalytic ability under solar light should not only be demonstrated experimentally, but should also be supported by characterization data.

4.  The “"bottom-up" assembly” and “spherical and flaky-shaped heterostructures” in the abstract are not reflected in the whole article, and is suggested to be modified.

5.     It is advisable to add the pure MoS2 and WO3 data to the XRD characterisation for comparison purposes.

6.    In morphological analysis section, the corresponding morphologies of MoS2 and WO3 should be indicated in separate circles and described in the article, with evidence of successful compounding of MoS2/WO3.

7.      In FTIR analysis, it is not only necessary to explain the origin and cause of the absorption peaks in the curve, but also the purpose of the FTIR characterisation of the sample and the conclusions obtained.

8.     It is desirable to add a UV-Vis diffuse reflectance absorption spectra of MoS2, WO3 and MoS2/WO3 to the UV analysis section to provide an indication of the origin of the band gap data as well as demonstrating the light absorption capability of the samples.

9.      In Figure 4, the curve labelled MoS2/WO3 sample yields band gap data for two pure MoS2, WO3. Please give a reasonable explanation and check for missing data.

10.   Logically, it would be preferable to exhibit Figure 6 first to demonstrate the degradation effect of the products and then Figure 5 to reflect the successful degradation of RhB by the products in RhB Photodegradation section.

11.  In Figure 6, the common order of presentation of the degradation graphs is: Irradiation time vs degradation (C/Co) curve - Degradation rate graph - kinetic graph - Cycles graph, which would make it more logical if arranged in this order.

12.   In Figure 6(b) for RhB Photodegradation, the curve is presented as a growing curve, which is usually used to show the output effect; the degradation curve is usually decaying, which is better to illustrate that the samples have degradation capability and made the concentration of RhB decreasing.

13.  The band gap values shown in Figure 7 are different from those calculated from Figure 4, please give a reasonable explanation and check for value mistakes.

14.   In the literatures of the same type, the conduction band value of MoS2 can be higher or lower than WO3. There are no characterizations or calculations of the CB value and VB value of MoS2 and WO3 in the article. Please provide data and explain the conclusion that the CB value of MoS2 is lower than that of WO3.

15.  What kind of mechanism is used in this paper to explain the transport of electrons from the conduction band of MoS2 with low potential energy to the conduction band of WO3 with high potential energy. The explanation in the text is not complete enough, somoreexplanationsareneeded. Although the products synthesized in the article are highly efficient in degrading RhB, there is too little data in the article to adequately explain the fundamental mechanisms that allow the products to degrade effectively under solar light. More data are needed to explain the improved performance of the product.

16.  Although the products synthesized in the article are highly efficient in degrading RhB, there is too little data in the article to adequately explain the fundamental mechanisms that allow the products to degrade effectively under solar light. More data are needed to explain the improved performance of the product.

17.    There is only one supplementary chart in the supplementary document, but it is numbered Figure 2. Please check whether there is data that has not been added in it.

18. Please make sure that the image numbering format is correct in the supplementary document.

Author Response

The paper describes a range of MoS2/WO3 photocatalysts that can efficiently degrade RhB under sunlight, and the motivation behind the problem studied in this manuscript is of interest. Nevertheless, there are a number of issues in this paper that need to be reconsidered after major revision. My detailed comments are as follows:

  1. The “Ex-situ Synthesis” in the title is not reflected in the whole article. If this is not the highlight of this article, we suggest deleting it.

Response: Thanks for your valuable suggestion. We synthesized nanostructures of MoS2 and WO3 by hydrothermal treatment and then prepared heterostructures of MoS2/WO3 by ex-situ synthesis instead of using the hydrothermal route again. Therefore, the word Ex-situ is justified and it has been further highlighted in the experimental part.

  1. The title of the article and the Materials and Methods section both indicate that the experiments in this paper were conducted under “Solar Light”, and it’s easy to make people confused that whether it’s real environmental sunlight or simulated sunlight. If it is real environmental sunlight, the light will be affected by weather or other conditions and therefore whether there is any control during the experiment. If not, it’d better to change the word to avoid confusion.

Response: Thanks for your valuable suggestion. The experiment was performed under Simulated Solar Simulator by Abet Technologies Sunlight TM Solar Simulator, and to avoid confusion word solar simulator has been used throughout the manuscript.

  1. The statement that the product has excellent catalytic ability under solar light should not only be demonstrated experimentally, but should also be supported by characterization data.

Response: Thanks for your valuable suggestion. The characteristics data has been provided in Figure 6.

  1. The “"bottom-up" assembly” and “spherical and flaky-shaped heterostructures” in the abstract are not reflected in the whole article, and is suggested to be modified.

Response: Thanks for your valuable suggestion. SEM analysis shows the flaky shape structures which were synthesized by the bottom-up technique for more evidence SEM figures have been provided in the Figure S2 of the supporting information.

  1. It is advisable to add the pure MoS2and WO3 data to the XRD characterization for comparison purposes.

Response: Thanks for your valuable suggestion and the XRD data has been provided in Figure S1 of the supporting information.

  1. In morphological analysis section, the corresponding morphologies of MoS2and WO3 should be indicated in separate circles and described in the article, with evidence of successful compounding of MoS2/WO3.

Response: Thanks for your valuable suggestion. Incorporated in Figure S2 of the supporting information.

  1. In FTIR analysis, it is not only necessary to explain the origin and cause of the absorption peaks in the curve, but also the purpose of the FTIR characterization of the sample and the conclusions obtained.

Response: Thanks for your valuable suggestion, In the discussion section of FTIR spectra, the data regarding the main compounds MoS2 and WO3 is reported. The peak functional groups like W=O, W-O, Mo-S-Mo, Mo-OH, S-O, etc were observed in FTIR spectra. Which indicates the presence of the heterostructure of MoS2/WO3.

  1. It is desirable to add UV-Vis diffuse reflectance absorption spectra of MoS2, WO3and MoS2/WO3 to the UV analysis section to provide an indication of the origin of the band gap data as well as demonstrating the light absorption capability of the samples.

Response: Thanks for your valuable suggestion, the band gap has been calculated by using Tauc's plot instead of the kubelka-Munk technique, therefore the UV-Vis absorption spectra have been provided in the Figure S4 of the supporting information.

  1. In Figure 4, the curve labelled MoS2/WO3sample yields band gap data for two pure MoS2, WO3. Please give a reasonable explanation and check for missing data.

Response: Thanks for your valuable suggestion. The band gaps calculated for MoS2 and WO3 have been provided in Figure S4 of the supporting information.

  1. Logically, it would be preferable to exhibit Figure 6first to demonstrate the degradation effect of the products and then Figure 5 to reflect the successful degradation of RhB by the products in RhB Photodegradation section.

Response: Thanks for the valuable suggestion, arrangements are correct as per conventional method.

  1. In Figure 6, the common order of presentation of the degradation graphs is: Irradiation time vs degradation (C/Co) curve - Degradation rate graph - kinetic graph - Cycles graph, which would make it more logical if arranged in this order.

Response: Thanks, it has been corrected as per suggestion.

  1. In Figure 6(b)for RhB Photodegradation, the curve is presented as a growing curve, which is usually used to show the output effect; the degradation curve is usually decaying, which is better to illustrate that the samples have degradation capability and made the concentration of RhB decreasing.

Response: Thanks, for photocatalytic degradation %-Efficiency curves have been shown instead of concentration, therefore the trend is increasing instead of decreasing. And the decreased concentration effect has been shown in Figure 6 of absorption curves.

  1. The band gap values shown in Figure 7are different from those calculated from Figure 4, please give a reasonable explanation and check for value mistakes.

Response: Figure 4 is showing the calculated bandgap whose absorption curves have been provided in the supporting information and Figure 7 is showing the absorption curves of RhB degradation.

  1. In the literatures of the same type, the conduction band value of MoS2can be higher or lower than WO3. There are no characterizations or calculations of the CB value and VB value of MoS2 and WO3 in the article. Please provide data and explain the conclusion that the CB value of MoS2 is lower than that of WO3.

Response: Thanks, for the bandgap potential Mott-Schottky-analysis is required which unfortunately cannot be done in Pakistan at such short notice, therefore we have proposed the possible photocatalytic mechanism following the literature.

  1. What kind of mechanism is used in this paper to explain the transport of electrons from the conduction band of MoS2with low potential energy to the conduction band of WO3 with high potential energy. The explanation in the text is not complete enough, so more explanations are  Although the products synthesized in the article are highly efficient in degrading RhB, there is too little data in the article to adequately explain the fundamental mechanisms that allow the products to degrade effectively under solar light. More data are needed to explain the improved performance of the product.

Response: Thanks for your valuable suggestion, Z-Scheme mechanism has been followed and more data has been added in the manuscript and the supporting information. The prepared heterostructures have shown high efficiency therefore, we have been working on more applications for the prospect.

  1. There is only one supplementary chart in the supplementary document, but it is numbered Figure 2. Please check whether there is data that has not been added in it.

Response: Thanks for your valuable suggestion, it has been extensively modified.

  1. Please make sure that the image numbering format is correct in the supplementary document.

Response: Thanks for your valuable suggestion it has been checked and modified.
